# DRGSPLAT: DEPTH-REGULARIZED 3D GAUSSIAN SPLATTING

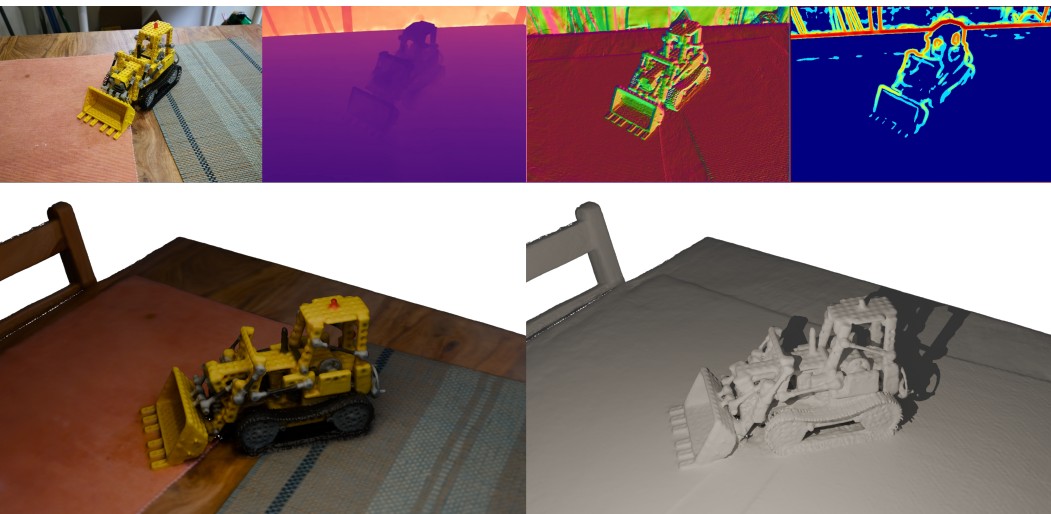

Figure 1: **DRGSplat** utilizes input images, monocular depth and normal predictions from an off-the-shelf network, and a depth-derived curvature map (visualized in this order from left to right) to regularize the rendered image and depth during 3DGS optimization, achieving accurate geometry and high-quality novel view synthesis without explicit constraints on the shape of the 3D Gaussians.

## ABSTRACT

3D Gaussian Splatting has rapidly become a leading method for photorealistic novel view synthesis. However, its geometric accuracy often lags behind its visual fidelity. Existing methods to improve geometry typically constrain the 3D Gaussians directly, compromising their volumetric nature. We introduce DRGSplat, a novel depth-regularization approach for 3D Gaussian Splatting that enhances geometric accuracy without direct modifications of the Gaussian primitives. DRGSplat regularizes the rendered depth maps during training with three key losses: a monocular depth loss enforcing global consistency, a surface normal loss refining local detail, and a new uncertainty-aware curvature loss that selectively penalizes high-gradient regions while avoiding the gradient instabilities common to direct curvature constraints. Experiments on standard benchmarks show that DRGSplat keeps the strong photometric quality of Gaussian Splatting while substantially improving geometric accuracy and outperforming state-of-the-art geometry-focused methods. On the ETH3D dataset, DRGSplat improves reconstruction accuracy, completeness, and F1 scores of 3DGS by 15, 25, and 17 percentage points, respectively. The source code will be made publicly available.

## 1 INTRODUCTION

Novel view synthesis (NVS) has seen tremendous progress in recent years, with applications ranging from virtual reality and augmented reality to robotics and autonomous driving. A key goal of NVS is to generate photorealistic images of a scene from arbitrary viewpoints, given a set of input images with known camera poses. Neural Radiance Fields (NeRF) (Mildenhall et al., 2021) have emerged as a powerful paradigm for NVS, representing scenes implicitly using a multi-layer perceptron (MLP) and achieving impressive rendering quality through volume rendering.

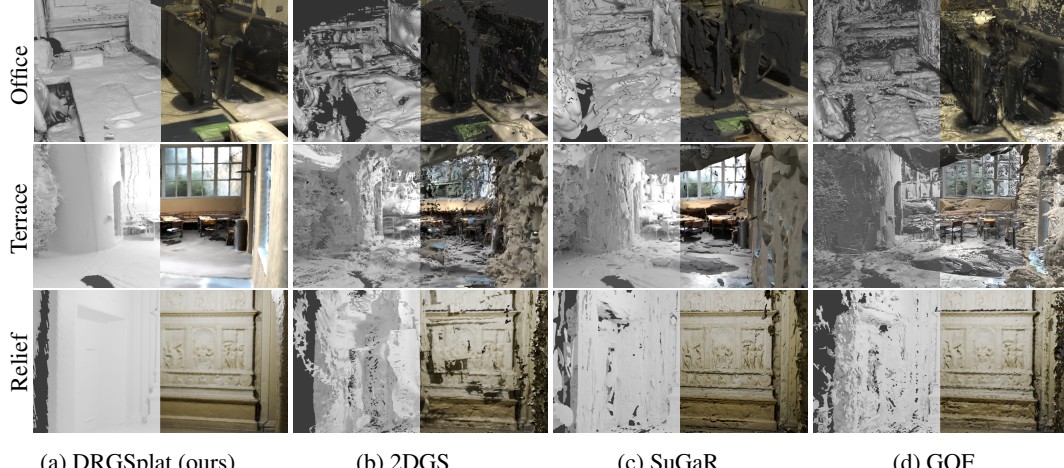

(a) DRGSplat (ours)      (b) 2DGS      (c) SuGaR      (d) GOF

Figure 2: **3D reconstructions** of our method, 2DGS (Zhu et al., 2023), SuGaR (Guédon & Lepetit, 2024), and GOF (Yu et al., 2024) on scenes from ETH3D (Schops et al., 2017).

Building upon the success of NeRF, 3D Gaussian Splatting (3DGS) (Kerbl et al., 2023) has recently been introduced as a highly efficient and effective alternative. 3DGS represents a scene as a collection of 3D Gaussian primitives rendered using differentiable rasterization. This approach enables real-time rendering while maintaining photorealistic quality, surpassing many NeRF-based methods in both speed and visual fidelity. However, while 3DGS excels at capturing the *appearance* of a scene, its *geometric accuracy* often lags largely behind its visual quality. This discrepancy between visual and geometric fidelity can be a significant limitation in applications where accurate 3D reconstruction is crucial, such as robotics, autonomous driving, and 3D modeling.

Existing methods aiming to improve the geometric accuracy of 3DGS often resort to directly constraining the shape or arrangement of the 3D Gaussian primitives themselves. For instance, some approaches enforce planarity constraints on the Gaussians (Zhu et al., 2023; Guédon & Lepetit, 2024), effectively "flattening" them to better align with surfaces. While this improves geometric accuracy in some cases, it fundamentally alters the nature of the Gaussian representation, limiting its ability to represent volumetric effects and potentially hindering the overall representational power of the model. This trade-off between geometric accuracy and volumetric representation is a key challenge in current 3DGS research.

In this paper, we introduce DRGSplat, a novel approach that addresses this challenge by regularizing the *rendered depth maps* produced during 3D Gaussian Splatting, rather than directly manipulating the Gaussian primitives. Our key insight is that accurate geometry can be encouraged indirectly by enforcing consistency between the rendered depth and readily available geometric priors, without sacrificing the volumetric nature of the Gaussians. Specifically, DRGSplat incorporates three complementary loss functions during the optimization process:

- **Monocular Depth Loss:** We leverage pre-trained monocular depth models (Hu et al., 2024) to provide geometric priors, enforcing consistency of rendered and predicted depth maps. This encourages the overall scene geometry to align with a plausible structure.
- **Monocular Normal Loss:** We incorporate pre-trained monocular normal estimation networks (Hu et al., 2024) to refine geometric details. By comparing the normals derived from the rendered depth with predicted normals, we encourage smooth and accurate surfaces.
- **Monocular Curvature Loss:** We introduce a novel curvature-based regularization term that further enhances geometric detail. Crucially, we incorporate an uncertainty estimation mechanism to selectively apply this regularization, focusing on regions where the estimates are reliable.

By combining these losses (see Fig. 1), DRGSplat achieves a significant improvement in geometric accuracy compared to vanilla 3DGS as well as to methods designed to enhance geometric fidelity, while maintaining comparable photometric quality. In summary, our contributions are: (i) A novel depth-regularization approach for 3DGS, via finite differences, that improves geometric accuracy without directly modifying the Gaussian primitives. (ii) A robust regularization approach incorporating monocular depth, monocular normals, and a novel uncertainty-aware curvature loss.

## 2 RELATED WORK

**Novel View Synthesis (NVS).** Significant advances have been made in NVS since the introduction of NeRF (Mildenhall et al., 2021). NeRF utilizes an MLP to represent scene geometry and appearance, achieving impressive rendering quality through volume rendering. Subsequent work has addressed various limitations. Mip-NeRF (Barron et al., 2021) and its successors (Barron et al., 2022; 2023) mitigate aliasing artifacts. Other efforts have focused on improving its efficiency through distillation (Reiser et al., 2021; Yu et al., 2021) and baking techniques (Hedman et al., 2021; Reiser et al., 2023; Xiangli et al., 2022). Feature grid-based representations (Chen et al., 2022; Hu et al., 2023; Fridovich-Keil et al., 2022; Müller et al., 2022; Sun et al., 2022) have also improved training and representational power. Most relevant to our work is 3D Gaussian Splatting (3DGS) (Kerbl et al., 2023), which achieves real-time rendering with high visual fidelity. While 3DGS is highly effective for NVS, its geometric accuracy is not its primary strength, which is the focus of our work.

**Geometric Accuracy in Neural Representations.** While NeRF and 3DGS excel at photorealism, improving geometric accuracy has been a separate research line. Some methods focus on incorporating geometric priors into NeRF, e.g., DS-NeRF (Deng et al., 2022) uses depth supervision. Other techniques, like NeuS (Wang et al., 2021), use normals. Several approaches (Zhu et al., 2023; Guédon & Lepetit, 2024; Turkulainen et al., 2024; Chung et al., 2024; Zhang et al., 2025) have specifically targeted improving the geometric fidelity of 3DGS. They directly constrain the Gaussians, often enforcing planarity or other priors. However, this can limit the representational power of the Gaussians, hindering their ability to capture 3D volumes. Our work takes a different approach by regularizing the rendered depth, avoiding direct manipulation of the Gaussian primitives.

**Monodepth and Normal Estimation.** Our method leverages monocular depth and normal estimation. Significant progress has been made in depth prediction using deep learning (Eigen et al., 2014; Laina et al., 2016; Fu et al., 2018; Ranftl et al., 2020; 2021; Hu et al., 2024; Chen et al., 2024; Yang et al., 2025). Similarly, normal estimation has advanced considerably (Bansal et al., 2016; Hu et al., 2024). We employ pre-trained networks for both tasks, demonstrating their effectiveness in regularizing 3DGS. Such monocular cues for regularization have been explored in prior works, such as MonoSDF (Yu et al., 2022) and DN-Splatter (Turkulainen et al., 2025). However, our work is the first to integrate them effectively within a 3DGS framework without imposing explicit constraints on the Gaussian primitives (e.g., enforcing flatness (Guédon & Lepetit, 2024; Zhu et al., 2023) or deriving ad-hoc normals directly from 3D Gaussians (Turkulainen et al., 2025; Yu et al., 2024)).

**Differentiable Rendering and Point-Based Graphics.** Our work is related to the broader field of differentiable rendering. Differentiable point-based rendering methods (Aliev et al., 2020; Insafutdinov & Dosovitskiy, 2018; Wiles et al., 2020; Yifan et al., 2019) offer efficiency and flexibility. Methods like NPBG (Aliev et al., 2020) and DSS (Yifan et al., 2019) rasterize point features, while Pulsar (Lassner & Zollhofer, 2021) introduces acceleration structures. 3DGS (Kerbl et al., 2023) itself falls into this category, using differentiable rasterization of 3D Gaussians. Our approach leverages the differentiability of the 3DGS rendering pipeline to incorporate our depth-based regularization.

**Concurrent Works.** Since the release of 3DGS (Kerbl et al., 2023), numerous works have proposed extensions and modifications. Some efforts (Gao et al., 2024; Bi et al., 2024) integrate normal information as additional Gaussian attributes, largely for relighting tasks. In contrast, our work directly targets geometric accuracy, achieving this through depth regularization rather than altering the Gaussian representation. Other approaches pursue more complex modifications: SuGaR (Guédon & Lepetit, 2024) approximates surfaces with 2D Gaussians, 2DGS (Zhu et al., 2023) replaces Gaussians with 2D disks, and NeuSG (Chen et al., 2023) combines them with an implicit SDF. Gaussian Opacity Fields (Yu et al., 2024) introduces new normal supervision and meshing pipelines, while DN-Splatter (Turkulainen et al., 2025) explores monodepth and normal supervision. In contrast, our DRGSplat offers a simpler and more principled solution: it improves geometric accuracy by leveraging monocular cues through an uncertainty-aware depth loss, without modifying Gaussian attributes. Instead of estimating flatness or normals from the 3D Gaussian parameters – which inherently lack well-defined normals – we compute them directly in image space via finite differences.

### 2.1 3D GAUSSIAN SPLATTING PRELIMINARIES

We build our method on 3DGS (Kerbl et al., 2023), a recent technique for NVS that represents a scene as a collection of anisotropic 3D Gaussian primitives. Unlike methods relying on implicit

representations (e.g., NeRF (Mildenhall et al., 2021)), 3DGS uses an explicit, differentiable representation, enabling efficient rendering. Each 3D Gaussian is defined by its mean position $\mathbf{p}_k \in \mathbb{R}^3$, a covariance matrix $\boldsymbol{\Sigma} \in \mathbb{R}^{3\times3}$, and an opacity $\alpha_k \in [0,1]$. The probability density function is as:

$$G(\mathbf{p}) = \exp\left(-\frac{1}{2}(\mathbf{p} - \mathbf{p}_k)^\top \boldsymbol{\Sigma}^{-1}(\mathbf{p} - \mathbf{p}_k)\right). \tag{1}$$

Covariance matrix $\boldsymbol{\Sigma}$ controls the shape and orientation of the Gaussian. For optimization, it is typically decomposed into a scaling matrix $\mathbf{S}$ and a rotation matrix $\mathbf{R}$, such that $\boldsymbol{\Sigma} = \mathbf{R}\mathbf{S}\mathbf{S}^\top\mathbf{R}^\top$. This decomposition allows for independent control over the size and orientation of each Gaussian.

To render an image, the 3D Gaussians are projected onto the 2D image plane. This involves transforming the Gaussians from the world coordinate system to the camera coordinate system using the world-to-camera transformation matrix $\mathbf{W}$, and then applying a local affine approximation of the projective transformation. This approximation, represented by the Jacobian matrix $\mathbf{J}$ of the projective transformation, allows for efficient computation. The covariance matrix in the camera space, $\boldsymbol{\Sigma}'$, is given by $\boldsymbol{\Sigma}' = \mathbf{J}\mathbf{W}\boldsymbol{\Sigma}\mathbf{W}^\top\mathbf{J}^\top$. The 2D covariance matrix $\boldsymbol{\Sigma}_{2D}$ is obtained by taking the upper-left $2 \times 2$ submatrix of $\boldsymbol{\Sigma}'$. This effectively projects the 3D Gaussian onto the image plane. The color of a pixel $\mathbf{x}$ is computed using alpha blending, accumulating contributions from all Gaussians that overlap the pixel, ordered from front to back as follows:

$$C(\mathbf{x}) = \sum_{k=1}^{K} c_k \alpha_k G_{2D_k}(\mathbf{x}) \prod_{j=1}^{k-1}(1 - \alpha_j G_{2D_j}(\mathbf{x})), \tag{2}$$

where $K$ is the number of Gaussians overlapping the pixel, $c_k$ is the view-dependent color of the $k$-th Gaussian (typically represented using spherical harmonics), $\alpha_k$ is its opacity, and $G_{2D_k}(\mathbf{x})$ is the 2D Gaussian probability density function evaluated at pixel location $\mathbf{x}$. The product term ensures that closer Gaussians occlude those farther away.

Similarly, the expected depth of a pixel $\mathbf{x}$ is computed as a weighted average of the depths of the contributing Gaussians, using the same alpha blending weights:

$$D(\mathbf{x}) = \sum_{k=1}^{K} d_k \alpha_k G_{2D_k}(\mathbf{x}) \prod_{j=1}^{k-1}(1 - \alpha_j G_{2D_j}(\mathbf{x})). \tag{3}$$

This depth rendering process is fully differentiable, allowing for gradients to be backpropagated to the Gaussian parameters.

## 3 DEPTH-REGULARIZED GAUSSIAN SPLATTING

### 3.1 MONOCULAR DEPTH LOSS

To enforce global geometric consistency, we incorporate a monocular depth loss that encourages the rendered depth $D(\mathbf{x})$ from the 3DGS to align with the depth predictions from a pre-trained monocular depth estimator following standard process. This leverages the ability of monocular networks to capture the overall scene structure even in the presence of ambiguities or lack of texture. However, monocular depth predictions typically provide depth up to an unknown scale and offset.

**Depth Alignment.** Following standard practice, we align the scale and shift of the predicted depth map $D_{predict}(\mathbf{x})$ to the rendered one $D(\mathbf{x})$. We leverage sparse depth observations from the COLMAP reconstruction, which provides accurate 3D point positions. Let $\mathcal{P}$ denote the set of pixels with available COLMAP depth values, and denote the depth at pixel $\mathbf{x} \in \mathcal{P}$ as $D_{colmap}(\mathbf{x})$. We solve for the optimal scale $s$ and shift $t$ by minimizing the L1 difference of the transformed predicted and COLMAP depths as follows:

$$s^*, t^* = \arg\min_{s,t} \sum_{\mathbf{x} \in \mathcal{P}} |s \cdot D_{predict}(\mathbf{x}) + t - D_{colmap}(\mathbf{x})|. \tag{4}$$

This robust linear regression problem can be efficiently solved using standard optimization methods (e.g., linear programming). The computed scale and shift are subsequently applied to the entire predicted depth map as $D'_{predict}(\mathbf{x}) = s^* \cdot D_{predict}(\mathbf{x}) + t^*$, yielding the aligned depth map $D'_{predict}(\mathbf{x})$.

**Depth Loss.** The monocular depth loss $\mathcal{L}_{depth}$ is defined as the confidence-weighted L1 difference of the rendered depth $D(\mathbf{x})$ and the aligned predicted depth $D'_{predict}(\mathbf{x})$ as follows:

$$\mathcal{L}_{depth} = \sum_{\mathbf{x}} c_d(\mathbf{x}) |D(\mathbf{x}) - D'_{predict}(\mathbf{x})|, \tag{5}$$

where the sum runs over all pixels $\mathbf{x}$. The per-pixel confidence $c_d(\mathbf{x}) \in [0, 1]$, obtained from the monodepth prediction network, modulates the construction of each pixel to the loss. This scheme effectively reduces the influence of uncertain or unreliable depth predictions, particularly in challenging areas such as textureless or reflective surfaces.

This simple yet powerful loss encourages the 3D Gaussians to produce depth maps consistent with monocular priors. Minimizing this loss guides the optimization towards geometrically plausible reconstructions, even in the absence of strong multi-view constraints. The differentiability of the depth rendering process ensures gradients can propagate effectively to update Gaussian parameters.

## 3.2 DIFFERENTIABLE NORMAL CALCULATION

Directly estimating or regularizing surface normals from 3D Gaussian primitives is inherently ambiguous, as Gaussians do not possess well-defined normals. Although each Gaussian has an associated orientation (defined by its covariance matrix), this orientation does *not* necessarily represent a surface normal. Instead, our key contribution is leveraging the rendered depth $D(\mathbf{x})$ to compute normals in a fully differentiable manner. This allows us to impose geometric constraints directly on the rendered depth rather than explicitly modifying Gaussian primitives. Our differentiable normal estimation procedure, inspired by classic finite difference methods, operates on the depth maps and enables end-to-end optimization of 3D Gaussian parameters.

**Finite Difference Approximation.** We approximate the partial derivatives of the depth map with respect to image coordinates $u$ and $v$ using Gaussian derivative kernels. Let $k_x$ and $k_y$ be 1D Gaussian derivative kernels of size $k$ and standard deviation $\sigma$, defined as:

$$k_x(i) = -\frac{i}{\sigma^2} \exp\left(-\frac{i^2}{2\sigma^2}\right), \quad k_y(j) = -\frac{j}{\sigma^2} \exp\left(-\frac{j^2}{2\sigma^2}\right), \tag{6}$$

where $i, j \in [-k//2, k//2]$. We then create 2D kernels, $k_{x_{2D}}$ and $k_{y_{2D}}$, through outer products. We consider the rendered depth $D(\mathbf{x})$, or the monocular depth $D_{mono}(\mathbf{x})$ as a 2D signal, and convolve it with these kernels using a 2D convolution operation with a stride of 1 and padding of $k//2$ to preserve the spatial dimensions. This yields the approximate partial derivatives $\frac{\partial D}{\partial u}$ and $\frac{\partial D}{\partial v}$ as:

$$\frac{\partial D}{\partial u} = \text{conv2d}(D, k_{x_{2D}}), \quad \frac{\partial D}{\partial v} = \text{conv2d}(D, k_{y_{2D}}). \tag{7}$$

**Normal Computation.** Given these partial derivatives, and assuming a camera model with depth aligned along the Z-axis, we approximate the normal $\mathbf{n}(\mathbf{x})$ at pixel location $\mathbf{x}$ as:

$$\mathbf{n}(\mathbf{x}) = \left[-\frac{\partial D}{\partial u}, -\frac{\partial D}{\partial v}, 1\right], \quad \hat{\mathbf{n}}(\mathbf{x}) = \mathbf{n}(\mathbf{x})/||\mathbf{n}(\mathbf{x})||. \tag{8}$$

The entire procedure is differentiable with respect to the depth map $D(\mathbf{x})$, and thus also differentiable with respect to the underlying Gaussian parameters.

Furthermore, we leverage the confidence scores (or uncertainties) provided by the monocular normal and depth estimator. Let $c_n(\mathbf{x}) \in [0, 1]$ represent the confidence of the predicted normal at pixel $\mathbf{x}$, and $c_d(\mathbf{x})$ represent the confidence of the depth. We use these confidences as per-pixel weights in the normal and depth-based loss functions, giving higher weight to predictions with higher confidence and reducing the influence of less reliable estimates. This allows for robust regularization.

**Normal Loss.** For the normal-based loss function, we borrow the formulation from MonoSDF (Yu et al., 2022). Specifically, the normal loss $\mathcal{L}_{normal}$ is defined as a weighted sum of an L1 loss and a cosine similarity loss of the rendered $\hat{\mathbf{n}}_{render}(\mathbf{x})$ and predicted normals $\hat{\mathbf{n}}_{predict}(\mathbf{x})$:

$$\mathcal{L}_{normal} = \sum_{\mathbf{x}} c_n(\mathbf{x})(\lambda_{L1}||\hat{\mathbf{n}}_{render}(\mathbf{x}) - \hat{\mathbf{n}}_{predict}(\mathbf{x})||_1 + \lambda_{cos}(1 - \hat{\mathbf{n}}_{render}(\mathbf{x}) \cdot \hat{\mathbf{n}}_{predict}(\mathbf{x}))),$$

where $\lambda_{L1}$ and $\lambda_{cos}$ are weighting factors controlling the relative importance of the L1 and cosine similarity terms, respectively. We use $\lambda_{L1} = \lambda_{cos} = 0.5$ in all experiments.

Figure 3: **Reconstructed meshes** with (top) and without color (bottom) by the proposed DRGSplat combined with vanilla 3DGS on scenes from different datasets.

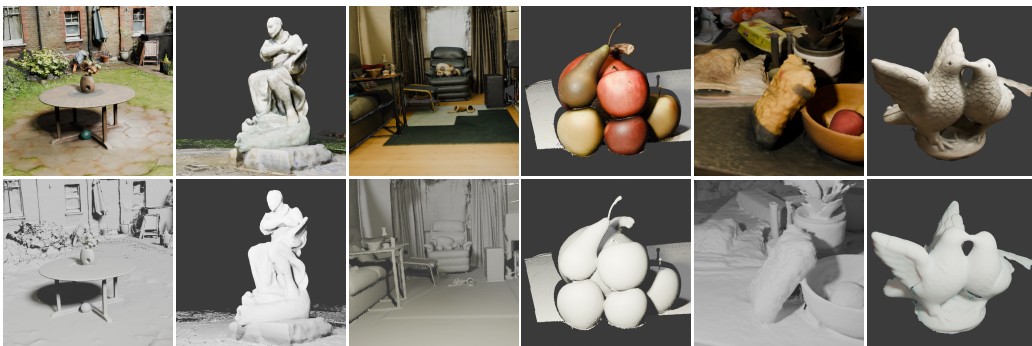

Table 1: **3D reconstruction** accuracy, completeness, and F1 scores at 5, 10, and 25 cm thresholds for 3DGS, SuGaR, 2DGS, DN-Splatter, Gaussian Opacity Fields (GOF), and our DRGSplat combined with 3DGS and GOF on the ETH3D dataset. Metrics are computed by comparing the reconstructed mesh to the ground truth scanned point cloud averaged over all scenes. The best results are in bold.

| Method | Acc @ 5cm ↑ | @ 10cm ↑ | @ 25cm ↑ | Comp @ 5cm ↑ | @ 10cm ↑ | @ 25cm ↑ | F1 @ 5cm ↑ | @ 10cm ↑ | @ 25cm ↑ |
|---|---|---|---|---|---|---|---|---|---|
| SuGaR | 10.8 | 19.4 | 37.8 | 18.8 | 32.2 | 59.5 | 13.2 | 23.5 | 45.3 |
| 2DGS | 17.7 | 30.4 | 52.9 | 21.6 | 35.3 | 60.3 | 18.0 | 30.6 | 53.5 |
| DN-Splatter | 16.7 | 27.4 | 50.1 | 34.5 | 47.9 | 76.9 | 23.6 | 36.9 | 62.1 |
| PGSR | 30.2 | 42.3 | 61.3 | 38.0 | 49.8 | 64.6 | 29.2 | 40.3 | 57.8 |
| 3DGS | 17.9 | 31.1 | 54.3 | 25.3 | 40.8 | 65.4 | 19.7 | 33.5 | 57.6 |
| + DRGSplat | **32.9** | **47.0** | **63.5** | **52.6** | **68.1** | **85.9** | **38.3** | **52.9** | **70.2** |
| GOF | 33.0 | 45.4 | 64.1 | 34.4 | 47.0 | 63.8 | 31.0 | 41.8 | 57.7 |
| + DRGSplat | **38.1** | **51.4** | **68.9** | **37.7** | **48.6** | **64.9** | **33.7** | **44.4** | **59.7** |
| COLMAP | 63.4 | 70.0 | 77.9 | 63.4 | 67.1 | 72.2 | 57.3 | 63.2 | 71.3 |

## 3.3 DIFFERENTIABLE CURVATURE CALCULATION

While normals provide directional constraints, explicitly constraining the rendered depth to match local curvature observations can enhance geometric consistency. We introduce a differentiable curvature measure derived from rendered depth maps, which complements normal-based regularization.

**Gradient Magnitude.** We first approximate the partial derivatives of the rendered depth $D(\mathbf{x})$ with respect to image coordinates using convolution with horizontal and vertical derivative kernels, similarly to Eq. 7. The gradient magnitude at pixel $\mathbf{x}$ is then defined as:

$$G(\mathbf{x}) = \sqrt{\left(\frac{\partial D}{\partial u}(\mathbf{x})\right)^2 + \left(\frac{\partial D}{\partial v}(\mathbf{x})\right)^2}, \qquad (9)$$

where a large $G(\mathbf{x})$ indicates high curvature, and $G(\mathbf{x}) \approx 0$ suggests flat regions. We employ a $5 \times 5$ Farid-Simoncelli kernel (Farid & Simoncelli, 2004) that we found best for curvature calculation.

**Curvature Measure.** We convert the gradient magnitude into a differentiable measure of local curvature as $C(\mathbf{x}) = 1/(1 + G(\mathbf{x}))$. High values of $C(\mathbf{x})$ correspond to nearly planar regions (low curvature), while lower values indicate high curvature or depth discontinuities.

**Curvature Confidence via Uncertainty Propagation.** Since the curvature $C(\mathbf{x})$ is derived from the gradient magnitude $G(\mathbf{x})$, which in turn is computed from finite differences of depth values, we propagate uncertainties from the monocular depth estimates accordingly. The monocular depth predictor provides, for each pixel $\mathbf{x}$, a depth mean $D(\mathbf{x})$ and an associated variance $\sigma_D^2(\mathbf{x})$, which we use directly as the depth uncertainty. Given the convolutional gradient operators in Eq. 7, the variance of each partial derivative can be approximated by

$$\sigma_{\frac{\partial D}{\partial u}}^2(\mathbf{x}) \approx \sum_{\mathbf{p}\in\Omega}\left(k_{x_{2D}}(\mathbf{p})\right)^2 \sigma_D^2(\mathbf{x}-\mathbf{p}), \quad \sigma_{\frac{\partial D}{\partial v}}^2(\mathbf{x}) \approx \sum_{\mathbf{p}\in\Omega}\left(k_{y_{2D}}(\mathbf{p})\right)^2 \sigma_D^2(\mathbf{x}-\mathbf{p}), \qquad (10)$$

Table 2: **Novel view synthesis results** on the ETH3D dataset (Schops et al., 2017).

| Metric | SuGaR | 2DGS | DN-Splatter | GOF | 3DGS | + DRGSplat |
|---|---|---|---|---|---|---|
| **PSNR** $\uparrow$ | 17.6 | 19.6 | 19.8 | 19.9 | 20.4 | **20.7** |
| **SSIM** $\uparrow$ | 0.585 | 0.725 | 0.745 | 0.709 | **0.774** | 0.764 |
| **LPIPS** $\downarrow$ | 0.471 | 0.360 | 0.322 | 0.409 | 0.317 | **0.315** |

where $\Omega$ is the kernel support. Assuming independence between horizontal and vertical gradients, these uncertainties are propagated to the gradient magnitude $G(\mathbf{x})$ as

$$\sigma_G^2(\mathbf{x}) \approx \frac{\left(\frac{\partial D}{\partial u}\right)^2 \sigma_{\frac{\partial D}{\partial u}}^2(\mathbf{x}) + \left(\frac{\partial D}{\partial v}\right)^2 \sigma_{\frac{\partial D}{\partial v}}^2(\mathbf{x})}{\left(\frac{\partial D}{\partial u}\right)^2 + \left(\frac{\partial D}{\partial v}\right)^2 + \epsilon}, \tag{11}$$

with $\epsilon > 0$ a small constant for numerical stability. The curvature confidence is then defined as $c_c(\mathbf{x}) = 1/(1 + \sigma_G^2(\mathbf{x}))$, which downweights pixels where the uncertainty in the depth gradients is high, while strongly enforcing curvature consistency in reliable regions.

**Curvature Loss.** Our curvature loss explicitly encourages local curvature consistency between the rendered depth map and the monocular curvature prior. Specifically, we define:

$$\mathcal{L}_{curvature} = \sum_{\mathbf{x}} c_c(\mathbf{x}) \Big| C(\mathbf{x}) - C_{\mathrm{predict}}(\mathbf{x}) \Big|, \tag{12}$$

where both $C(\mathbf{x})$ and $C_{\mathrm{predict}}(\mathbf{x})$ are obtained from the depth maps, as described earlier. Since $C(\mathbf{x})$ is differentiable with respect to rendered depth $D(\mathbf{x})$, and thus the underlying Gaussian parameters, this loss provides gradients that guide optimization toward locally consistent curvature.

## 4 EXPERIMENTS

**Implementation details.** We implement the proposed losses into the 3DGS code in NerfStudio (Tancik et al., 2023). We use the default parameters in all experiments. Also, we include the proposed losses into Gaussian Opacity Fields (Yu et al., 2024). All calculations are done via vectorized operations on a GPU. The time for calculating the normals from rendered depth takes on average 13.9 ms for an image of size $1550 \times 1034$. The flatness map is calculated in 5.1 ms. For predicting depth and normals, we use Metric3D v2 (L) (Hu et al., 2024).

**Mesh Extraction.** To evaluate geometric accuracy and compare with mesh-based methods, we extract a triangle mesh following the procedure from (Zhu et al., 2023). We render depth maps that are then fused using the TSDF integration in Open3D (Zhou et al., 2018), with a voxel size of 0.004 and a truncation threshold of 0.02 (Zhu et al., 2023). A triangle mesh is extracted from the TSDF volume using Marching Cubes (Lorensen & Cline, 1998). We observed that this process sometimes produces small, "floating" artifacts in the mesh. To mitigate this, we perform a filtering step, removing connected components with fewer than $n = 50$ triangles.

**Datasets.** We evaluate our method on three standard real-world datasets. We use the *ETH3D* (Schops et al., 2017), *DTU* (Jensen et al., 2014), and *Tanks and Temples* (Knapitsch et al., 2017) datasets to assess the geometric fidelity of our reconstructions. ETH3D consists of diverse indoor and outdoor scenes with high-quality ground-truth depth and precise camera calibration, enabling rigorous quantitative evaluation. We follow the standard evaluation protocol, measuring reconstruction accuracy, completeness, and F1 score against the ground-truth scan. The dataset includes 13 scenes from the training split of the stereo dataset, with images downsampled to a quarter of their original resolution. We use every eighth frame as a test set to compute photometric losses. For both DTU and Tanks and Temples, we follow the standard protocol (Zhu et al., 2023).

*Mip-NeRF360 Dataset:* To ensure that our geometric improvements do not compromise photometric quality, we evaluate our method on the Mip-NeRF360 dataset (Barron et al., 2022). This dataset contains $360°$ real-world scenes with varying scales and intricate details, designed to test rendering fidelity. We train our model on the official training split and evaluate it on the corresponding test views, reporting standard photometric metrics (e.g., PSNR, SSIM, and LPIPS).

**Baselines.** To evaluate the effectiveness of our approach, we compare DRGSplat with five state-of-the-art Gaussian Splatting-based methods: 3DGS, 2DGS, SuGaR, DN-Splatter, and GOF.

Table 3: **Quantitative comparison on the DTU Dataset** (Jensen et al., 2014). We report the Chamfer distance. The results of the baselines are taken from GOF (Yu et al., 2024). The proposed DRGSplat significantly improves both 3DGS and GOF.

| Method | 24 | 37 | 40 | 55 | 63 | 65 | 69 | 83 | 97 | 105 | 106 | 110 | 114 | 118 | 122 | Mean |
|---|---|---|---|---|---|---|---|---|---|---|---|---|---|---|---|---|
| NeRF | 1.90 | 1.60 | 1.85 | 0.58 | 2.28 | 1.27 | 1.47 | 1.67 | 2.05 | 1.07 | 0.88 | 2.53 | 1.06 | 1.15 | 0.96 | 1.49 |
| VolSDF | 1.14 | 1.26 | 0.81 | 0.49 | 1.25 | 0.70 | 0.72 | 1.29 | 1.18 | 0.70 | 0.66 | 1.08 | 0.42 | 0.61 | 0.55 | 0.86 |
| NeuS | 1.00 | 1.37 | 0.93 | 0.43 | 1.10 | 0.65 | 0.57 | 1.48 | 1.09 | 0.83 | 0.52 | 1.20 | 0.35 | 0.49 | 0.54 | 0.84 |
| Neuralangelo | 0.37 | 0.72 | 0.35 | 0.35 | 0.87 | 0.54 | 0.53 | 1.29 | 0.97 | 0.73 | 0.47 | 0.74 | 0.32 | 0.41 | 0.43 | 0.61 |
| SuGaR | 1.47 | 1.33 | 1.13 | 0.61 | 2.25 | 1.71 | 1.15 | 1.63 | 1.62 | 1.07 | 0.79 | 2.45 | 0.98 | 0.88 | 0.79 | 1.33 |
| GaussianSurfels | 0.66 | 0.93 | 0.54 | 0.41 | 1.06 | 1.14 | 0.85 | 1.29 | 1.53 | 0.79 | 0.82 | 1.58 | 0.45 | 0.66 | 0.53 | 0.88 |
| 2DGS | 0.48 | 0.91 | 0.39 | 0.39 | 1.01 | 0.83 | 0.81 | 1.36 | 1.27 | 0.76 | 0.70 | 1.40 | 0.40 | 0.76 | 0.52 | 0.80 |
| 3DGS | 2.14 | 1.53 | 2.08 | 1.68 | 3.49 | 2.21 | 1.43 | 2.07 | 2.22 | 1.75 | 1.79 | 2.55 | 1.53 | 1.52 | 1.50 | 1.96 |
| **+ DRGSplat** | **0.92** | **1.34** | **0.57** | **0.67** | **1.43** | **0.86** | **0.76** | **1.43** | **1.44** | **0.92** | **1.09** | **1.50** | **0.66** | **0.73** | **0.64** | **1.00** |
| GOF | 0.50 | 0.82 | 0.37 | 0.37 | 1.12 | 0.74 | 0.73 | 1.18 | 1.29 | 0.68 | 0.77 | 0.90 | 0.42 | 0.66 | 0.49 | 0.74 |
| **+ DRGSplat** | **0.41** | **0.60** | **0.31** | **0.35** | **0.81** | **0.69** | **0.65** | **1.14** | **1.13** | **0.67** | **0.58** | **0.77** | **0.41** | 0.67 | **0.46** | **0.64** |

Table 4: **Novel view synthesis results** on outdoor and indoor scenes from the Mip360 dataset (Barron et al., 2022). Baseline results are from (Zhu et al., 2023).

| | Outdoor Scenes | | | Indoor Scenes | | |
|---|---|---|---|---|---|---|
| Method | PSNR ↑ | SSIM ↑ | LPIPS ↓ | PSNR ↑ | SSIM ↑ | LPIPS ↓ |
| SuGaR | 22.93 | 0.629 | 0.356 | 29.43 | 0.906 | 0.225 |
| 2DGS | 24.34 | 0.717 | 0.246 | 30.40 | 0.916 | 0.195 |
| GOF | 24.82 | 0.750 | **0.202** | 30.79 | 0.924 | 0.184 |
| 3DGS | 24.64 | 0.731 | 0.234 | 30.41 | 0.920 | 0.189 |
| **+ DRGSplat** | **26.44** | **0.765** | 0.228 | 30.46 | **0.929** | **0.107** |

*SuGaR* (Guédon & Lepetit, 2024) integrates explicit surface regularization, constraining the Gaussians to align with an underlying surface prior. 2DGS (Zhu et al., 2023) models the scene as a set of 2D discs with properties similar to those of the Gaussians to align better with surfaces. DN-Splatter (Turkulainen et al., 2025) uses a depth regularization strategy similar to ours. We apply DN-Splatter using the same Metric3Dv2 depth predictions as in our method. GOF (Yu et al., 2024) uses depth and normal supervision and marching tetrahedra to obtain a 3D mesh.

**Metrics.** We evaluate our method using standard geometric metrics: accuracy, completeness, and F1 score, following the ETH3D protocol. Accuracy measures the average distance from reconstructed points to the closest GT points, capturing how well the reconstruction aligns with the reference. Completeness quantifies the coverage of the reconstruction by computing the average distance from GT points to their nearest reconstructed ones. High accuracy indicates precise reconstruction, while high completeness suggests minimal missing data. To balance both aspects, we also report the F1 score, which computes the harmonic mean of accuracy and completeness. We evaluate these metrics at three distance thresholds (5, 10, and 25 cm) to assess both fine-grained details and broader structural correctness. For novel view synthesis, we report the usual PSNR, SSIM, and LPIPS scores.

**Experiments on ETH3D.** We evaluate geometric accuracy on ETH3D by comparing the reconstructed mesh to the ground-truth scan. Table 1 presents a quantitative comparison with state-of-the-art methods. 3DGS + DRGSplat significantly outperforms all baselines in F1 score. On average, DRGSplat improves accuracy by 0.3 percentage points, completeness by over 20.5 points, and F1 score by 10.3 points compared to the closest competitor, GOF. These results highlight the effectiveness of our depth regularization approach in supervising 3D Gaussian Splatting. The proposed DRGSplat also improves GOF. However, 3DGS + DRGSplat leads to better results on this dataset. We show example reconstructions in Fig. 2. The proposed method leads to significantly more detailed, complete, and smoother results than all baselines in all examples.

For reference, we also include results from COLMAP (Schonberger & Frahm, 2016), a state-of-the-art multi-view stereo (MVS) method, shown in the last row of Table 1. While Gaussian Splatting-based reconstructions still fall short of MVS in terms of accuracy, DRGSplat significantly narrows the gap. Moreover, our method achieves superior completeness scores at 10 cm and 25 cm thresholds, demonstrating the advantages of integrating monocular cues into 3D Gaussian Splatting.

To ensure that these geometric improvements do not compromise photometric quality, we also evaluate using standard photometric metrics (PSNR, SSIM, and LPIPS). The results in Table 2 confirm that DRGSplat maintains or improves photometric accuracy, demonstrating that enhanced geometry does not come at the cost of rendering fidelity.

Table 5: **Quantitative comparison on the Tanks and Temples dataset** (Knapitsch et al., 2017). We report the F1 score. The results of the baselines are taken from GOF (Yu et al., 2024). The proposed DRGSplat significantly improves both 3DGS and GOF.

| Method | Barn | Meetingroom | Ignatius | Courthouse | Caterpillar | Truck | Mean |
|--------|------|-------------|----------|------------|-------------|-------|------|
| SuGaR | 0.14 | 0.15 | 0.33 | 0.08 | 0.15 | 0.26 | 0.19 |
| 2DGS | 0.41 | 0.17 | 0.51 | 0.16 | 0.23 | 0.45 | 0.32 |
| VCR-Gaussian | 0.62 | 0.19 | 0.61 | 0.19 | 0.26 | 0.52 | 0.40 |
| 3DGS | 0.13 | 0.01 | 0.04 | 0.09 | 0.08 | 0.19 | 0.09 |
| **+ DRGSplat** | **0.43** | **0.34** | **0.51** | **0.28** | **0.31** | **0.47** | **0.39** |
| GOF | 0.51 | **0.28** | 0.68 | 0.28 | 0.41 | 0.59 | 0.46 |
| **+ DRGSplat** | **0.54** | **0.28** | **0.69** | **0.34** | **0.45** | **0.62** | **0.49** |

Table 6: **Ablation study on the proposed losses** on the ETH3D dataset (Schops et al., 2017), demonstrating the impact of the depth ($\mathcal{L}_d$), normal ($\mathcal{L}_n$), and curvature losses ($\mathcal{L}_c$). We report accuracy (Acc), completeness (Comp), and F1 scores at thresholds 5, 10 and 25 cm.

| Method | Acc (%) ↑ | | | Comp (%) ↑ | | | F1 (%) ↑ | | |
|--------|-----------|------|------|------------|------|------|----------|------|------|
| | @5cm | @10cm | @25cm | @5cm | @10cm | @25cm | @5cm | @10cm | @25cm |
| 3DGS | 17.9 | 31.1 | 54.3 | 25.3 | 40.8 | 65.4 | 19.7 | 33.5 | 57.6 |
| $+ \mathcal{L}_d$ | 30.3 | 44.8 | 62.7 | 35.2 | 49.1 | 69.3 | 31.3 | 45.4 | 64.7 |
| $+ \mathcal{L}_d + \mathcal{L}_n$ | 32.4 | 46.7 | **64.5** | 48.1 | 63.5 | 83.9 | 36.0 | 50.5 | **70.2** |
| $+ \mathcal{L}_d + \mathcal{L}_n + \mathcal{L}_c$ | **32.9** | **47.0** | 63.5 | **52.6** | **68.1** | **85.9** | **38.3** | **52.9** | **70.2** |

**Experiments on DTU.** We further assess geometric quality on the DTU benchmark, focusing on reconstructing small objects instead of full scenes. Table 5 presents a comparison with state-of-the-art methods, including both implicit surface representations and other 3DGS techniques. DRGSplat provides a significant improvement to the geometric accuracy of standard 3D Gaussian Splatting. Applying our regularization reduces the mean Chamfer distance from 1.96 to 1.00, an improvement of nearly 50%. This demonstrates that our approach successfully addresses one of the key weaknesses of the original 3DGS method. Furthermore, DRGSplat is complementary to existing geometry-focused methods. When combined with GOF, we reduce the mean Chamfer error by 13.5%, from 0.74 to 0.64, leading to the best results achieved by 3DGS alternatives.

**Experiments on Tanks and Temples.** Table 5 reports results on the Tanks and Temples benchmark Knapitsch et al. (2017). DRGSplat yields consistent improvements over both 3DGS and GOF across all scenes. When applied to 3DGS, the average F1 score increases from 0.09 to 0.39, with large gains on all scenes. When combined with GOF, the mean score rises from 0.46 to 0.49, with improvements on Barn, Courthouse, Caterpillar, and Truck. These results demonstrate that the proposed depth-based regularization scales well to complex outdoor environments and benefits both lightweight and stronger Gaussian pipelines.

**Experiments on Mip-NeRF360.** To further validate our approach, we evaluate DRGSplat on the Mip-NeRF360 dataset using PSNR, SSIM, and LPIPS. Table 4 presents the results. The results of the baselines were copied from (Zhu et al., 2023). DRGSplat achieves consistent improvements over baseline methods, notably improving PSNR by 1.8 points compared to the second-best approach, 3DGS. Additionally, our method demonstrates improvements across SSIM and LPIPS, outperforming baselines in nearly all cases. These results confirm that our geometric enhancements do not degrade photometric quality but instead contribute to improved rendering performance.

**Ablation Studies.** To analyze the impact of our proposed losses, we conduct an ablation study on ETH3D, shown in Table 6. We report accuracy, completeness, and F1 score for different loss configurations, progressively incorporating each component into the standard 3DGS optimization. The last row, 3DGS with all proposed losses, corresponds to DRGSplat. Our findings show that the monodepth loss has the largest impact, nearly doubling the accuracy compared to standard 3DGS. The normal loss further refines accuracy, and it contributes to improving completeness by reinforcing local surface consistency. The curvature loss provides an additional large boost in completeness, improving scores by 4-5 percentage points while maintaining high accuracy. These results demonstrate that each loss plays a complementary role, collectively leading to substantial improvements.

Visualizations using different loss configurations are provided in Fig. 4, showing how each term progressively removes surface artifacts. The monodepth loss corrects large-scale geometric distor-

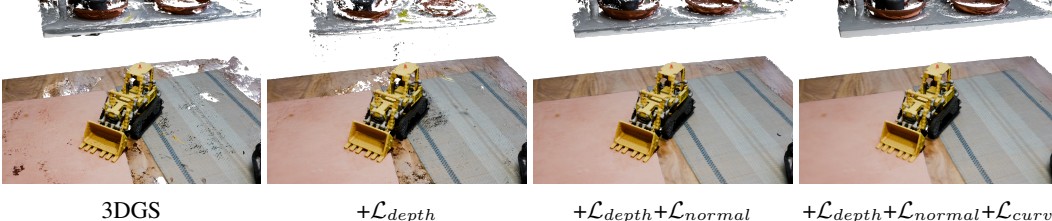

| 3DGS | $+\mathcal{L}_{depth}$ | $+\mathcal{L}_{depth}+\mathcal{L}_{normal}$ | $+\mathcal{L}_{depth}+\mathcal{L}_{normal}+\mathcal{L}_{curv}$ |

Figure 4: **3D reconstructions** of the *kitchen* scene from the Mip-360 dataset. We show the mesh produced by standard 3DGS and then add our losses column-by-column, each further improving geometric quality. With all proposed losses included, the tabletop becomes flat, and the white background table with the plant pots exhibits the most coherent and recognizable structure.

Table 7: **Ablation study on the depth predictor** on the ETH3D dataset (Schops et al., 2017). We report the F1 scores at thresholds 5, 10, 25 cm.

| Method | | @ 5cm | @ 10cm | @ 25cm |
|---|---|---|---|---|
| Metric3d v2 (S) | (Hu et al., 2024) | 32.1 | 48,0 | 68.7 |
| Metric3d v2 (L) | (Hu et al., 2024) | 35.1 | 49.1 | 68.9 |
| Metric3d v2 (G) | (Hu et al., 2024) | 38.3 | 52.9 | 70.2 |
| MoGe (L) | (Wang et al., 2025) | **42.1** | **55.5** | 74.7 |
| Depth Anything v2 (L) | (Yang et al., 2025) | 33.6 | 48.9 | **76.6** |

tions, the normal loss sharpens local structure, and the curvature loss suppresses high-frequency irregularities, producing the smooth and coherent surfaces seen in the final reconstruction.

Table 7 examines the influence of different monocular depth predictors. As expected, using smaller Metric3D variants leads to reduced F1 scores due to the drop in depth quality. We also evaluate Depth Anything v2 (Yang et al., 2025), which does not provide normals, requiring us to estimate them via finite differences. While this model performs well at coarse thresholds, its accuracy degrades at finer ones, likely due to the absence of an explicit normal prediction. In addition, we run our method using depth from the recent MoGe Wang et al. (2025). Because MoGe does not output depth confidence, we approximate it using the normalized gradient magnitude computed with a Sobel filter, which effectively marks depth as uncertain near image edges. The results show that MoGe can further improve our method, outperforming Metric3Dv2 on this benchmark.

**Runtimes.** On the ETH3D dataset, using an NVIDIA GeForce RTX 4090, standard 3DGS requires approximately 7 minutes per scene. Our 3DGS + DRGSplat variant runs in about 16 minutes and additionally requires a single forward pass of the monocular depth network for each image, with Metric3Dv2 taking roughly 50-100 ms per image. Although this introduces a moderate overhead relative to vanilla 3DGS, the overall cost remains substantially lower than that of other geometry-optimizing approaches. For example, PGSR and Gaussian Opacity Fields require an average of 131 and 91 minutes, respectively, on ETH3D while producing lower geometric accuracy.

## 5 CONCLUSION

We introduced DRGSplat, a novel depth-regularized approach for 3D Gaussian Splatting that significantly improves geometric accuracy while preserving rendering quality. Importantly, we show that state-of-the-art geometric accuracy is achievable without directly constraining Gaussian primitives. By incorporating monocular depth, normal priors, and curvature loss, our method refines the scene geometry without directly constraining the 3D Gaussian parameters. Experiments on ETH3D and DTU demonstrate that DRGSplat outperforms the state of the art, improving accuracy by 0.3, completeness by 20.5, and F1 score by 10.3 percentage points. Our method also narrows the gap with MVS approaches like COLMAP, particularly in completeness at larger thresholds. Evaluations on Mip-NeRF360 confirm that our geometric improvements do not degrade photometric quality, achieving 1.8 dB higher PSNR than 3DGS. Overall, our results show that monocular priors and geometric constraints can significantly enhance 3DGS, leading to state-of-the-art geometry and novel view synthesis. Importantly, this improvement is achieved without explicitly constraining the shape of Gaussian primitives. The source code will be publicly released.

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
