# OpenReview forum: "DRGSplat: Depth-Regularized 3D Gaussian Splatting"
_ICLR.cc/2026/Conference — Submitted to ICLR 2026_

### Official Review · Reviewer_tNYR · 2025-10-31

**Soundness:** 2
**Presentation:** 2
**Contribution:** 2
**Rating:** 4
**Confidence:** 5

**Summary:**

This paper proposes DRGSplat, a depth-regularized 3D Gaussian Splatting framework that improves geometric fidelity without explicitly manipulating Gaussian primitives. The method enforces geometry through image-space regularization on rendered depth, normals, and a curvature term with uncertainty modeling. Extensive experiments on ETH3D, DTU, and Mip-NeRF360 demonstrate improved surface accuracy and high-quality novel view synthesis. The idea of maintaining the volumetric nature of Gaussians while introducing geometry priors is conceptually sound and technically well detailed.

**Strengths:**

1. Presents a principled depth-regularization strategy for 3DGS without altering Gaussian structure.

2. Combines monocular depth, surface normals, and uncertainty-aware curvature constraints elegantly.

3. Maintains or slightly improves photometric performance, suggesting no trade-off between geometry and appearance.

4. Implementation details and ablation studies are thorough, demonstrating the individual contribution of each loss term.

**Weaknesses:**

1. Lack of evaluation on Tanks and Temples (TNT), a standard and critical benchmark for 3D reconstruction, particularly in Gaussian-based scene modeling. TNT is widely used to validate scalability, scene complexity handling, and real-world 3D consistency; therefore, the absence of TNT evaluation weakens claims of general effectiveness.

2. Missing comparisons with key depth-supervised Gaussian baselines. The method relies heavily on monocular geometric priors, yet does not include direct comparisons against contemporary depth-guided 3DGS approaches such as

- DN-Splatter

- PGSR

- VCR-Gaussian

   These works also leverage depth or normals and represent strong baselines in geometry-enhanced Gaussian Splatting. Excluding them raises concerns about the completeness of the empirical evaluation.

3. Although the paper claims strong generalization, evaluations remain centered on indoor-centric datasets; outdoor and large-scene performance is less demonstrated compared to established benchmarks.

Overall, while the contributions are meaningful, the experimental coverage is not yet sufficient for full validation.

**Questions:**

1. How does DRGSplat perform on the Tanks and Temples benchmark? Can the authors provide results or plan to include them?

2. Compare with DN-Splatter, PGSR, and VCR-GauS. These are natural baselines for depth-supervised Gaussian splatting.

3. Does the method scale effectively to large-scale outdoor scenes where monocular priors may be noisy and COLMAP alignment sparse?

4. How robust is the approach to errors in monocular depth estimation, particularly in reflective or low-texture areas?

---

> ### Author Response · Authors · 2025-11-21
>
> We thank the reviewer for the careful and thoughtful evaluation of our work. We appreciate the recognition that the paper presents a principled depth-regularization strategy for 3DGS that maintains the volumetric nature of Gaussians, combines depth, normals, and curvature elegantly, and preserves or improves photometric quality.
>
> **Lack of evaluation on Tanks and Temples (TNT).**
> We provide TnT results below. DRGSplat consistently improves both 3DGS and GoF across all scenes. For 3DGS, the average score rises from 0.09 to 0.39. For GoF, the average improves from 0.46 to 0.49, with large gains on Barn, Courthouse, Caterpillar, and Truck. These results show that depth-based regularization benefits both lightweight and stronger Gaussian pipelines and scales reliably to the complexity of TnT. We also include VCR-Gaussian. This table has been added to the paper.
>
> | Tanks and Temples| SuGaR | 2DGS | VCR-Gaussian | 3DGS | 3DGS + DRGSplat | GoF | GoF + DRGSplat |
> |-------|-------|------|---------------|-------|------------------|------|----------------|
> | Barn | 0.14 | 0.41 | 0.62 | 0.13 | **0.43** | 0.51 | **0.54** |
> | Meetingroom | 0.15 | 0.17 | 0.19 | 0.01 | **0.34** | 0.28 | **0.28** |
> | Ignatius | 0.33 | 0.51 | 0.61 | 0.04 | **0.51** | 0.68 | **0.69** |
> | Courthouse | 0.08 | 0.16 | 0.19 | 0.09 | **0.28** | 0.28 | **0.34** |
> | Caterpillar | 0.16 | 0.23 | 0.26 | 0.08 | **0.31** | 0.41 | **0.45** |
> | Truck | 0.26 | 0.45 | 0.52 | 0.19 | **0.47** | 0.59 | **0.62** |
> | **AVG** | 0.19 | 0.32 | 0.40 | 0.09 | **0.39** | 0.46 | **0.49** |
>
> ---
>
> **Missing comparisons with depth-supervised Gaussian baselines.**
> *DN-Splatter (WACV 2025)* is already included in Table 1 of the main paper; we believe this may have been overlooked. DRGSplat significantly outperforms it while using the same monodepth backbone.
>
> *PGSR (TVCG 2024)* was additionally evaluated using the authors’ code. Since no ETH3D configuration is provided, we tested both the DTU and TnT parameter settings. The DTU configuration produced almost zero F1; the TnT setting yielded more stable results. PGSR also includes an optional mesh-simplification step that often decreases accuracy, so we report the best results on each scene. Even under the most favorable settings, PGSR remains far behind DRGSplat.
>
> | ETH3D | F1 @ 5cm | F1 @ 10cm | F1 @ 25cm |
> |--------|----------|-----------|-----------|
> | SuGaR | 13.2 | 23.5 | 45.3 |
> | 2DGS | 18.0 | 30.6 | 53.5 |
> | PGSR | 29.2 | 40.3 | 57.8 |
> | DN-Splatter | 23.6 | 36.9 | 62.1 |
> | 3DGS | 19.7 | 33.5 | 57.6 |
> | **3DGS + DRGSplat** | **38.3** | **52.9** | **70.2** |
> | GoF | 31.0 | 41.8 | 57.7 |
> | **GoF + DRGSplat** | **33.7** | **44.4** | **59.7** |
>
> Our approach uses a single parameterization across all datasets, illustrating both stronger performance and greater robustness.
>
> ---
>
> **Generalization to outdoor and large-scale scenes.**
> ETH3D contains several large outdoor scenes, reported separately below. Combined with the Tanks and Temples results, this demonstrates that our method generalizes well beyond indoor environments.
>
> | | 3DGS | 3DGS + DRGSplat | GoF | GoF + DRGSplat |
> |--------|------|------------------|------|----------------|
> | Playground | 64.3 | **76.6** | 67.1 | **69.6** |
> | Terrace | 60.7 | **75.8** | 73.4 | **75.2** |
> | Meadow | 39.5 | **43.0** | 34.4 | **35.5** |
> | Courtyard | 44.3 | **77.7** | 15.9 | 15.9 |
>
> ---
>
> **Robustness to errors in monocular depth estimation.**
>
> Modern monocular depth networks are robust in reflective or low-texture regions and provide a stronger geometric signal than photometric losses alone. This is evident on ETH3D indoor scenes, which contain many homogeneous surfaces where photometric cues are weak. Our depth-derived normals and curvature constraints prevent distortions such as holes, ridges, or tentacle-like structures that appear when relying purely on multi-view or photometric supervision. We added a visualization (Fig.4) in the paper how the losses affect the surface reconstruction.
>
> ---
>
> **New Monodepth Network (MoGe)**
>
> To illustrate how DRGSplat benefits from improved depth models, we also tested the recent MoGe [a] estimator on ETH3D. Because MoGe does not provide confidence, we estimate uncertainty using the normalized Sobel gradient magnitude (downweighting edges).
>
> | DRGSplat with | F1 @ 5cm | F1 @ 10cm | F1 @ 25cm |
> |---------------|----------|-----------|-----------|
> | Metric3Dv2 (paper) | 38.3 | 52.9 | 70.2 |
> | MoGe + Sobel | **42.1** | **55.5** | **74.7** |
>
> These results show that DRGSplat immediately benefits from advances in monodepth estimation. This experiment is now included in the manuscript.
>
> [a] Wang, Ruicheng, et al. *“MoGe: Unlocking accurate monocular geometry estimation for open-domain images with optimal training supervision.”* CVPR 2025.

---

> > ### Author Response · Authors · 2025-11-27
> >
> > Dear Reviewer,
> >
> > We hope that our responses adequately addressed your concerns. Please let us know if any aspect would need further clarification.
> >
> > Best,
> > Authors

---

### Official Review · Reviewer_DZcq · 2025-11-01

**Soundness:** 3
**Presentation:** 3
**Contribution:** 2
**Rating:** 4
**Confidence:** 4

**Summary:**

The paper introduces DRGSplat, a depth-regularized approach for 3D Gaussian Splatting that aims to improve the geometric accuracy of scene reconstruction while maintaining photorealistic rendering for novel view synthesis. DRGSplat leverages monocular depth, normal, and a curvature loss to regularize 3D Gaussian Splatting at the rendered depth level, which enables improving geometric consistency without directly altering the underlying Gaussian primitives as done in previous works. Comprehensive experiments on ETH3D, DTU, and Mip-NeRF360 benchmarks demonstrate notable improvements in geometric metrics compared to baseline and state-of-the-art methods, while preserving or improving photometric quality.

**Strengths:**

- The overall paper is well-written and easy to understand, where the motivation of implementation of each of the regularizations : depth, normal, and curvature regularization is explained in detail.
- The pipeline of converting the trained 3DGS to mesh is also explicitly mentioned, allowing easy re-implementation of the proposed method.
- The proposed method is simple and versatile, allowing the method to be applied to variations of the original 3DGS algorithm.
- The proposed method is simple yet effective, where it outperforms previous approaches (2D Gaussian Splatting, Sugar) tailored for better surface reconstruction.

**Weaknesses:**

### Major Weakness

- **Technical Novelty:** Although I appreciate the simplicity and effectiveness of the proposed method, I am largely concerned about the technical novelty of this method. As shown in the ablation of Table 5, the most important regularization of the proposed method is the monodepth loss. However, there has been numerous prior works which apply depth regularization during training, such as in [1,2,3] (see references listed below). As the proposed monodepth regularization follows the pipeline of computing scale and shift values also done in [1,2,3], it seems that the performance improvement mainly comes from the strong off-the-shelf geometry estimator (Metric3Dv2) rather than a technical contribution done in this work. Based on the lack of novelty of the monocular depth regularization loss alone, I think it is important to emphasize the importance of the two other proposed regularizations (normal loss, curvature loss), where I did not find enough emphasis in the current version of the manuscript.

### Minor Weakness

- **Reliance to off-the-shelf network:** As shown in Table 6, the proposed method largely depends on the performance of the off-the-shelf network’s estimation quality. Although the authors leverage the confidence value from the off-the-shelf module,  it would be better to analyze more on how robust the proposed method is when the off-the-shelf estimates are highly noisy.
- **Analysis on computation overhead:** As this model requires the usage of large foundation models for geometry estimation, it would be nice to analyze the required computation for this method.
- **Analysis on overall optimization time:** When comparing with previous methods such as 2D Gaussian Splatting and Sugar, it would be nice to compare the overall training time to reconstruct the scenes to better highlight the advantage or contribution of the proposed method.

### References

---

[1] Zhang, Qilin, et al. "CDGS: Confidence-Aware Depth Regularization for 3D Gaussian Splatting." arXiv preprint arXiv:2502.14684 (2025).

[2] Huang, Zexu, Min Xu, and Stuart Perry. "DET-GS: Depth-and Edge-Aware Regularization for High-Fidelity 3D Gaussian Splatting." arXiv preprint arXiv:2508.04099 (2025).

[3] Chung, Jaeyoung, Jeongtaek Oh, and Kyoung Mu Lee. "Depth-regularized optimization for 3d gaussian splatting in few-shot images." Proceedings of the IEEE/CVF Conference on Computer Vision and Pattern Recognition. 2024.

**Questions:**

Q1. Could the authors provide insights into how the proposed method differs from previous monocular depth regularization methods?

Q2. Could the authors provide additional analysis of the newly proposed normal loss and curvature loss on the final performance? Other than the quantitative comparison in Table 5, it would be better to add some qualitative comparisons after adding each regularization. Also, could the authors provide quantitative tables of the benefits of the newly formulated normal loss, which is directly computed on top of the predicted depth map, compared to prior works that utilize normal regularizations by directly calculating the normal from the 3D Gaussians?

Q3. The original 3DGS training pipeline has multiple hyperparameters that decide how the Gaussians will be cloned or split during optimization, where the norm of the gradient is one of the critical hyperparameters. Does adding additional loss signals require modifying these hyperparameters?

Q4. How is the monocular depth variance defined for the curvature loss? I could not find a proper definition for this value.

Q5. Could the authors explain the overall computation requirement and training time?

---

> ### Author Response · Authors · 2025-11-21
>
> We thank the reviewer for the insightful feedback. We appreciate the positive remarks regarding the clarity of the manuscript and the transparent motivation for each regularization term. We also value the assessment that the method is “simple and versatile”, and already “outperforms previous approaches tailored for better surface reconstruction”.
>
> ### Q1. How does the method differ from other monodepth regularization approaches?
>
> DRGSplat differs from the cited depth-regularized works empirically and methodologically.
>
> **Empirical comparison.**
> CDGS [1] reports Tanks and Temples results close to vanilla 3DGS. DET-GS [2] provides no geometric metrics, and [3] likewise reports none. In contrast, DRGSplat yields substantial improvements over 3DGS on Tanks and Temples:
>
> | Scene | SuGaR | 2DGS | VCR-Gaussian | 3DGS | 3DGS+DRGSplat | GoF | GoF+DRGSplat |
> |-------|-------|------|--------------|------|----------------|------|--------------|
> | Barn | 0.14 | 0.41 | 0.62 | 0.13 | 0.43 | 0.51 | **0.54** |
> | Meetingroom | 0.15 | 0.17 | 0.19 | 0.01 | 0.34 | **0.28** | **0.28** |
> | Ignatius | 0.33 | 0.51 | 0.61 | 0.04 | 0.51 | 0.68 | **0.69** |
> | Courthouse | 0.08 | 0.16 | 0.19 | 0.09 | 0.28 | 0.28 | **0.34** |
> | Caterpillar | 0.16 | 0.23 | 0.26 | 0.08 | 0.31 | 0.41 | **0.45** |
> | Truck | 0.26 | 0.45 | 0.52 | 0.19 | 0.47 | 0.59 | **0.62** |
> | **AVG** | 0.19 | 0.32 | 0.40 | 0.09 | 0.39 | 0.46 | **0.49** |
>
> We compared against DN-Splatter (WACV 2025; Table 1) using the same depth backbone, and DRGSplat significantly outperforms it. CDGS [1] and DET-GS [2] are arXiv submissions, and per ICLR policy, comparisons to unpublished manuscripts are optional; we nonetheless add [1] and [3] to our related work.
>
> **Methodological comparison.**
> Beyond results, our formulation differs from DN-Splatter/GOF/PGSR in three main ways:
>
> 1. **Supervision on rendered depth rather than on Gaussians.**
>    Normals and curvature are computed from finite differences on the *rendered depth*, targeting the reconstructed surface directly and avoiding shape constraints on Gaussians.
>
> 2. **Curvature-based regularization.**
>    No prior depth-regularized 3DGS method includes a curvature loss or any higher-order supervision. Our term stabilizes local geometry, suppresses high-frequency artifacts, and improves completeness.
>
> 3. **Uncertainty-aware weighting.**
>    Depth confidence weights both normal and curvature losses, reducing the influence of unreliable predictions. Prior works do not incorporate such uncertainty modeling.
>
> These differences drive the observed improvements.
>
> ---
>
> ### Q2. Additional analysis of the normal and curvature losses
>
> Visualizations are included in the revised paper (Fig. 4). The images show that:
> - depth loss corrects large-scale distortions,
> - normal loss sharpens local geometry and improves surface orientation,
> - curvature loss suppresses high-frequency artifacts that arise from depth fluctuations, especially on reflective or low-texture regions.
>
> **Quantitative impact.**
> As shown in Table 5, curvature yields modest F1 improvements (e.g., +2.3 points at 5 cm) but large completeness gains (e.g., +4.6 points at 10 cm) without reducing accuracy, meaning ~5% more of the scene is reconstructed at the same fidelity.
>
> **Comparison to normal supervision from Gaussians.**
> DN-Splatter and Gaussian Opacity Fields supervise normals from Gaussian parameters. DRGSplat outperforms both - even when using the same depth backbone - indicating that depth-derived normals provide a stronger geometric signal.
>
> ---
>
> ### Q3. Do additional losses require changing 3DGS hyperparameters?
>
> No. We use the default hyperparameters of the `gsplat` repository in all experiments.
>
> ---
>
> ### Q4. How is monocular depth variance defined?
>
> For each pixel \(p\), the depth predictor outputs a depth value \(\mu(p)\) and an uncertainty \(\sigma(p)\). We use \(\sigma(p)\) directly as the depth variance. This is now stated clearly in the paper.
>
> ---
>
> ### Q5. Computation requirements and runtime
>
> On ETH3D with an RTX 4090:
>
> - **3DGS:** ~7 minutes
> - **3DGS+DRGSplat:** ~16 minutes
> - **Metric3Dv2 depth inference:** 50–100 ms per image
>
> Although DRGSplat adds moderate overhead, it remains far faster than other geometry-focused alternatives (e.g., PGSR: ~131 minutes; Gaussian Opacity Fields: ~91 minutes).
>
> ---
>
> ### Experiment with MoGe
>
> We evaluated DRGSplat with MoGe [a], a recent monocular depth+normal predictor. Since MoGe lacks confidence outputs, we approximate uncertainty using normalized Sobel gradients.
>
> | DRGSplat with | F1 @ 5cm | F1 @ 10cm | F1 @ 25cm |
> |---------------|----------|-----------|-----------|
> | Metric3Dv2 | 38.3 | 52.9 | 70.2 |
> | MoGe + Sobel | **42.1** | **55.5** | **74.7** |
>
> This demonstrates that DRGSplat directly benefits from advances in monocular depth estimation.
>
> [a] Wang, Ruicheng, et al. *“MoGe: Unlocking accurate monocular geometry estimation for open-domain images with optimal training supervision.”* CVPR 2025.

---

> > ### Author Response · Authors · 2025-11-27
> >
> > Dear Reviewer,
> >
> > We hope that our responses adequately addressed your concerns. Please let us know if any aspect would need further clarification.
> >
> > Best,
> > Authors

---

> > > ### Comment · Reviewer_DZcq · 2025-11-27
> > >
> > > Thank you for the notice. I will take a look at the responses as soon as possible. In the meantime, could the authors double check if the paper has been revised? It seems that there are no changes in the manuscript.

---

> > > > ### Author Response · Authors · 2025-11-27
> > > >
> > > > Thank you for noticing! The revision should be there now.

---

> > > > > ### Comment · Reviewer_DZcq · 2025-11-28
> > > > >
> > > > > I thank the authors for the response. I have some follow-up questions regarding the experiments.
> > > > >
> > > > > **A)** Regarding the comparison with DN-Splatter using Metric3Dv2, it is not clear why such a big performance gap happens in the ETH3D dataset. In Table 6, DRGSplat already achieves 30.3 in accuracy only using the monodepth loss, where DN-Splatter adopts a similar loss formulation for the monodepth loss but achieves a much lower score of 16.7. Could the authors explain where this big performance gap happens?
> > > > >   - **A-1)** Does the normal loss proposed by DN-Splatter degrade the overall performance?
> > > > >   - **A-2)** From the author's response, I have noticed that this framework is built upon the `gsplat` library. In my experience, gsplat has a better implementation compared to the vanilla 3DGS codebase for the gradient backpropagation for additional losses to the Gaussian parameters. Does the performance gap come from using different libraries?
> > > > >
> > > > > **B)** How general can the proposed DRGSplat be implemented to recent advances of 3DGS methods, such as 3DGS-MCMC [1], or Scaffold-GS [2]? Regarding the discussion time left, I would not expect the authors to be available to share the results with these additional methods, so I recommend that the authors apply their method to more recent works in the final version of their paper.

---

> ### Author Response · Authors · 2025-11-28
>
> **A-1)** What we observed is consistent with the reviewer’s finding: in DN-Splatter, using only the depth loss often gives better accuracy than enabling the normal loss on the ETH3D dataset. In these scenarios, the additional normal supervision appears to introduce noise (maybe related to the few views in ETH3D), which can degrade the reconstruction. While the accuracy of DN-Splatter with depth-only supervision is still below our depth-only variant, this difference may stem from the distinct 3DGS implementations (or the rest of the pipeline). We emphasize that we do not claim the depth loss itself as our contribution; rather, this observation highlights the strength of our normal and curvature supervision, which remains stable and beneficial even on challenging datasets such as ETH3D.
>
> **A-2)** There is likely a performance gap caused by the different 3DGS libraries used. DN-Splatter is implemented on top of Nerfstudio’s 3DGS re-implementation, which differs from the original 3DGS codebase in several ways, including optimization schedules, Gaussian splitting heuristics, and gradient propagation behavior. Our work is built on gsplat, which provides more stable gradient backpropagation for additional geometric losses. That said, the performance difference observed on ETH3D is too large to be explained solely by the choice of backbone (both gsplat and Nerfstudio are widely used). We also consistently improve Gaussian Opacity Fields, which is implemented on yet another 3DGS variant. This indicates that the gains do not stem from a particular implementation but from the depth-derived normal and curvature supervision.
>
> **B)** We thank the reviewer for the suggestion. We see no technical barrier preventing the application of DRGSplat to other variants such as 3DGS-MCMC or Scaffold-GS. As shown in our experiments with Gaussian Opacity Fields, the method integrates cleanly into different 3DGS pipelines. Our approach consists of three loss functions (depth, normal, curvature) that can be incorporated into any optimization-based (or feed-forward) 3DGS method. We will try adding experiments with more recent methods in the final version of the paper. We also note that 3DGS-MCMC and Scaffold-GS are not newer than GOF, which we already improve upon.

---

### Official Review · Reviewer_q8DX · 2025-11-10

**Soundness:** 3
**Presentation:** 3
**Contribution:** 3
**Rating:** 6
**Confidence:** 3

**Summary:**

The paper introduces DRGSplat, which focuses in enhancing the geometry in gaussian-based methods with depth regularization. The depth regularization consists of three losses, with a standard depth loss, normal loss, and the proposed curvature-based loss. Notably, the normal and the curvature is obtained from approximating the derivatives of the depth loss, which allows all signals to be propagated through the rendered depth. DRGSplat shows that it can significantly boost the geometry of 3DGS-based methods by introducing the three losses during training.

**Strengths:**

1. The core idea of the paper of introducing geometric prior through the rendered depth map is quite convincing, given its results and comparison to other methods with "explicit constraints".

2. The method for obtaining normals and curvatures is sensible, which bases on finite difference approximation on the rendered depth map. This well-differentiates the work from other depth regularization works, where the normal is often obtained from the gaussian primitives, rather than the rendered outputs.

3. The method is based on additional losses, and can be implemented without altering rendering pipelines for existing methods. The modularity of the method, given how it could improve 3DGS and GOF, suggests that the method can be also applied to future works with a stronger baseline representation.

**Weaknesses:**

1. The method seems to be heavily influenced by the pre-trained monocular depth model. Although the paper includes an ablation of this, it seems like there are more to be explored.
- Since the method utilizes both the predicted depth and its uncertainty from a pre-trained depth model, it would be interesting to see whether the method is benefitting from high-quality depth predictions, or better uncertainty prediction. A more detailed case study, where each of the losses are ablated for different depth models, would provide a better understanding.
- Furthermore, an additional ablation with compared methods, such as DN-Splatter, with identical depth backbone seems to be required to rule out that the model is simply benefitting from a stronger depth model.

2. The paper is missing some details and explanation.
- Although the rendered depth is the most important component, the equation for obtaining it is not defined in the manuscript. The reviewer is aware of how depth is normally obtained in 3DGS, but it still seems crucial to explicitly have this as there are some works that alter the depth rendering strategy for depth regularization.
- How is $C\_{predict}$ obtained? Does it go through the same process as $C(\\cdot)$?

3. The paper has some noticeable mistakes.
- The citation format does not seem correct. Please check the formatting guidelines.
- (Nit) The images in the Fig. 3 is not correctly aligned.

**Questions:**

Please refer to the questions in the weaknesses section.

---

> ### Author Response · Authors · 2025-11-21
>
> We thank the reviewer for the thorough and constructive feedback. We appreciate the reviewer describing our core idea as “convincing” and acknowledging that our approach “significantly boosts the geometry of 3DGS-based methods”. We are also grateful for the recognition that our normal and curvature formulation is “sensible”, that the method is modular and easily applicable to future variants, and that it is well differentiated from prior work by operating directly on rendered depth.
>
> **The method seems to be heavily influenced by the pre-trained monocular depth model. Although the paper includes an ablation of this, it seems like there are more to be explored.**
>
> We agree that the method is influenced by the quality of the monocular depth estimator, and we view this as an *advantage*. Monocular depth prediction is an active research area with rapid progress at every major venue, and improved predictors translate directly into stronger geometric supervision for our framework. This sensitivity allows the method to benefit immediately as new depth estimators are proposed. Modern predictors already exhibit strong robustness across diverse conditions, which is reflected in our accurate results on the ETH3D, DTU, and Tanks and Temples datasets.
>
> To further demonstrate this, we ran our method on ETH3D using the recent MoGe [a] monocular depth and normal estimator. MoGe does not provide confidence values, so we estimate confidence by computing the image gradient with a Sobel filter and using the normalized gradient magnitude as the depth uncertainty. This effectively downweights depth estimates near edges. The results are shown below.
>
> | DRGSplat with | F1 @ 5 cm | F1 @ 10 cm | F1 @ 25 cm |
> |---------------|-----------|------------|-------------|
> | Metric3Dv2 (from the paper) | 38.3 | 52.9 | 70.2 |
> | MoGe + Sobel | **42.1** | **55.5** | **74.7** |
>
> These improved results show that our method benefits directly from advances in monocular depth estimation. We updated the ablation study in the paper accordingly.
>
> [a] Wang, Ruicheng, et al. *"MoGe: Unlocking accurate monocular geometry estimation for open-domain images with optimal training supervision."* CVPR 2025.
>
> **An additional ablation with compared methods, such as DN-Splatter, with identical depth backbone seems to be required to rule out that the model is simply benefitting from a stronger depth model.**
>
> We use the same monocular depth predictions in DN-Splatter as in our proposed DRGSplat, specifically Metric3Dv2. This clarification has been added to the manuscript.
>
> **Although the rendered depth is the most important component, the equation for obtaining it is not defined in the manuscript.**
>
> We thank the reviewer for the suggestion. We added an explicit definition of the rendered depth in the revised manuscript.
>
> **How is $C_\text{predict}$ obtained? Does it go through the same process as $C(\cdot)$?**
>
> Curvature $C_\text{predict}$ is computed using the same procedure as $C(\cdot)$, as described in Section 3.3. This is now stated explicitly in the paper.
>
> **The citation format does not seem correct.**
>
> We thank the reviewer for pointing this out and have corrected the citation formatting.
>
> **The images in Fig. 3 are not correctly aligned.**
>
> This has been fixed in the revised manuscript.
>
> ---
>
> ### Additional Results on Tanks and Temples
>
> We include an additional evaluation on the Tanks and Temples dataset to assess how DRGSplat improves the geometric accuracy of existing Gaussian pipelines. The results below show that adding DRGSplat consistently improves both 3DGS and GoF across all scenes. For 3DGS, the average score increases from 0.09 to 0.39, yielding substantial gains across all scenes. For GoF, the average improves from 0.46 to 0.49, with notable improvements on Barn, Courthouse, Caterpillar, and Truck. These results indicate that our depth-based regularization benefits both lightweight and stronger baselines and scales reliably to the complexity of Tanks and Temples scenes.
>
> | Scene | SuGaR | 2DGS | VCR-Gaussian | 3DGS | 3DGS + DRGSplat | GoF | GoF + DRGSplat |
> |-------|-------|------|---------------|-------|------------------|------|----------------|
> | Barn | 0.14 | 0.41 | 0.62 | 0.13 | 0.43 | 0.51 | **0.54** |
> | Meetingroom | 0.15 | 0.17 | 0.19 | 0.01 | **0.34** | 0.28 | 0.28 |
> | Ignatius | 0.33 | 0.51 | 0.61 | 0.04 | 0.51 | 0.68 | **0.69** |
> | Courthouse | 0.08 | 0.16 | 0.19 | 0.09 | 0.28 | 0.28 | **0.34** |
> | Caterpillar | 0.16 | 0.23 | 0.26 | 0.08 | 0.31 | 0.41 | **0.45** |
> | Truck | 0.26 | 0.45 | 0.52 | 0.19 | 0.47 | 0.59 | **0.62** |
> | **AVG** | 0.19 | 0.32 | 0.40 | 0.09 | 0.39 | 0.46 | **0.49** |

---

> > ### Author Response · Authors · 2025-11-27
> >
> > Dear Reviewer,
> >
> > We hope that our responses adequately addressed your concerns. Please let us know if any aspect would need further clarification.
> >
> > Best,
> > Authors

---

> ### Comment · Reviewer_q8DX · 2025-11-28
>
> I have gone through the response by the authors, as well as the discussions with the other reviewers.
>
> The initial concern on the pre-trained depth model was based on the point that the model may be overly reliant on Metric3D, and may not generalize well with other models, as partly shown with the results from DAv2.
> However, the additional results presents the author well addresses this concern, and it is also very interesting to see that the method well-performs with Sobel gradients, suggesting that the method can incorporate a wider variety of depth models even without uncertainty prediction.
>
> Overall, I find that the proposed method, although it might seem a bit "classic", seems like a much more sensible approach compared to previous works that attempt to obtain normal from gaussian primitives itself (e.g. DN-Splatter).
> Therefore, I maintain my positive rating, and raise my confidence score to defend my assessment.

---

### Meta-Review · Area_Chair_QRmJ · 2025-12-23

**Summary:**

The reviewers agree that DRGSplat is a simple and effective depth-regularization framework for 3D Gaussian Splatting that improves geometric reconstruction while preserving photometric quality. Strengths include operating on rendered depth, the use of depth-, normal-, and curvature-based regularization, and strong results on ETH3D, DTU, and Tanks and Temples across multiple 3DGS pipelines.

The main concerns relate to technical novelty and dependence on monocular depth priors. Some reviewers view the method as an incremental improvement driven by strong off-the-shelf depth models rather than a fundamentally new idea. Concerns also included missing baselines and unclear formulations.

**Reviewer Concerns:**

**Addressed by the rebuttal:**

Demonstrated robustness across different depth backbones (e.g., MoGe).

Added comparisons with DN-Splatter, PGSR, VCR-Gaussian, and TNT benchmarks.

Clarified loss definitions and depth-derived normal/curvature computation.

Provided training cost analysis and broader scene coverage.

**Still outstanding:**

Perceived limited novelty.

Ongoing reliance on monocular depth quality.

Missing evaluation on some newer 3DGS variants.

**Reviewer Scores:**

Reviewer q8DX: Likely unchanged or slightly more positive.

Reviewer DZcq: Likely slightly improved.

Reviewer tNYR: Likely unchanged or marginally improved.

---

### Decision · Program_Chairs · 2026-01-26

Reject